# Practical Application of a Urinary Zearalenone Monitoring System for Feed Hygiene Management of a Japanese Black Cattle Breeding Herd—Relevance to Anti-Müllerian Hormone and Serum Amyloid A Clarified from a Two-Year Survey

**DOI:** 10.3390/toxins15050317

**Published:** 2023-04-30

**Authors:** Oky Setyo Widodo, Dhidhi Pambudi, Makoto Etoh, Emiko Kokushi, Seiichi Uno, Osamu Yamato, Masayasu Taniguchi, Mirni Lamid, Mitsuhiro Takagi

**Affiliations:** 1Joint Graduate School of Veterinary Sciences, Yamaguchi University, Yamaguchi 753-8515, Japan; oky.widodo@fkh.unair.ac.id (O.S.W.); masa0810@yamaguchi-u.ac.jp (M.T.); 2Division of Animal Husbandry, Faculty of Veterinary Medicine, Airlangga University, Surabaya 60115, Indonesia; osam@vet.kagoshima-u.ac.jp (O.Y.); mirnylamid@fkh.unair.ac.id (M.L.); 3Department of Mathematics Education, Faculty of Teacher Training and Education, Sebelas Maret University, Surakarta 57126, Indonesia; dhidhipambudi@staff.uns.ac.id; 4Ohita Agricultural Mutual Aid Association, Taketa 878-0024, Japan; mako_eto@nosai-oita.jp; 5Faculty of Fisheries, Kagoshima University, Kagoshima 890-0056, Japan; kokushi@fish.kagoshima-u.ac.jp (E.K.); uno@fish.kagoshima-u.ac.jp (S.U.); 6Joint Faculty of Veterinary Medicine, Kagoshima University, Kagoshima 890-0065, Japan; 7Laboratory of Theriogenology, Joint Faculty of Veterinary Medicine, Yamaguchi University, Yamaguchi 753-8515, Japan

**Keywords:** cattle, urine, zearalenone, AMH, SAA, long-term monitoring, calving interval

## Abstract

In this study, a herd of Japanese Black (JB) breeding cattle with sporadic reproductive disorders was continuously monitored for an additional year to assess the effects of the urinary zearalenone (ZEN) concentration and changes in parameters (AMH and SAA) with time-lag variables and herd fertility (reproductive performance). This herd had high (exceeded the Japanese dietary feed regulations) urinary ZEN and rice straw ZEN concentrations (1.34 mg/kg). Long-term data of the herd with positive ZEN exposure revealed a decreasing ZEN concentration in urine and a gradual decrease in the AMH level with age. The AMH level was significantly affected by the ZEN value 2 months earlier and the AMH level in the previous month. The changes in ZEN and SAA values were significantly affected by the ZEN and SAA values in the previous month. Additionally, calving interval data between pre-monitoring and post-monitoring showed a significantly different pattern. Furthermore, the calving interval became significantly shorter between the time of contamination (2019) and the end of the monitoring period (2022). In conclusion, the urinary ZEN monitoring system may be a valuable practical tool for screening and detecting herd contamination in the field, and acute and/or chronic ZEN contamination in dietary feeds may affect herd productivity and the fertility of breeding cows.

## 1. Introduction

Zearalenone (ZEN), an estrogenic mycotoxin from *Fusarium* species that can disrupt cattle reproductive physiology, is one of the most commonly identified mycotoxins in grains and animal feed [1,2]. Several studies reported mycotoxin contamination in feed by detecting the presence of ZEN [3,4,5,6,7]. Despite efforts to minimize mycotoxin concentrations in animal feeds, the prevalence of *Fusarium*-derived mycotoxins is increasing worldwide. ZEN is not degraded by feed processing such as milling, extrusion, storage, and heating (thermally stable) [8,9]. Therefore, monitoring and detecting ZEN contamination is important [10]. The adverse effects of mycotoxins on reproductive performance should be minimized by managing and caring for livestock to avoid mycotoxin contamination. Zearalenone contamination in herds can be assessed by measuring urinary ZEN metabolites in selected cows because it is absorbed in the gastrointestinal tract and excreted in urine after hepatic metabolism. Previously, we established a urinary ZEN monitoring system in cattle using ELISA for initial screening purposes, followed by LC-MS/MS validation to detect ZEN and its related metabolites [11,12,13]. We also evaluated the efficacy of mycotoxin adsorbents (MAs) added to the diet to reduce the intestinal uptake of mycotoxins [13].

Building on our previously published study [11], we applied our monitoring system on Japanese Black (JB) breeding cattle herds and reported the usefulness of the system to detect a ZEN-contaminated herd, and the possible relationship among the parameters such as ZEN, anti-Müllerian hormone (AMH; a potential predictor of fertility, superovulation response, ovarian disorders, and herd longevity [14,15,16,17]), and serum amyloid A (SAA; known as an acute-phase protein, and a sensitive and not only early indicator of inflammation but also an effective diagnostic aid in animal reproduction [18,19,20]) concentrations based on their monthly measurements of fixed five cows during the 1-year follow-up ZEN contamination mitigation period under a ZEN-affected herd. As emphasized in our previous report [11], we believe that long-term monitoring of spontaneous (naturally occurring) cases is key to clarifying the effects of dietary mycotoxin contamination on cattle health, especially the reproductive efficacy of breeding cattle herds. We continued monitoring for an additional year after the last report [11], providing monthly guidance on feeding management to farmers and confirming the effectiveness of countermeasures based on the results of two years of follow-up.

In this communication, we report the follow-up results obtained from our long-term urinary ZEN monitoring system in a JB cattle herd from July 2020 to July 2022. Here, we analyzed the effects of long-term (2 years) relationships among the ZEN-AMH-SAA associations obtained from our monthly on-farm follow-up measurements, and how the reproductive efficacies improved following our consultation with the farmers based on the results of the monthly ZEN-AMH-SAA measurements, whose relationship could not be clarified from only the one-year results alone, but was clarified using the two-year data in this herd.

## 2. Results

### 2.1. Long-Term (2 Years) Monitoring of ZEN in a JB Breeding Cattle Herd

We sampled urine and blood monthly for two years, from July 2020 to July 2022. The data for the first half of the year have been previously reported [11]. In this communication, the data for the second half of the year and those over the two years of study are reported.

We used a monthly sample of five cows in this study. In February 2022, a new cow was added to replace another cow that would be culled in the following month. Thus, the total number of cows in that month was six. From March 2022 to the end of this study, the number of cows returned to normal (five cows), and a new cow from the same herd was included. Monthly changes in both urinary ZEN and serum AMH concentrations are shown in Figure 1. Previously, it was reported that there were two peaks of urinary ZEN contamination detected during monitoring in August 2020 and April 2021 [11]. Subsequently, the herd’s urinary ZEN contamination was maintained at a low and steady level. Overall, monitoring of urinary ZEN contamination from July 2020 to July 2022 revealed a declining trend, as indicated by the red dotted line. Similar to the urinary ZEN concentration, the serum AMH level showed a decreasing trend, as indicated by the blue dotted lines. The mean, highest, and lowest AMH values ± SEM were 1431.4 ± 39.07, 1820.8 ± 228.14, and 1083 ± 176.41 pg/mL, respectively.

The results of the vector autoregression (VAR) model between ZEN, AMH, and SAA values (second-year data) and ZEN, AMH, and SAA changes (second-year data) are shown in Table 1 and Table 2. The results of the VAR model between ZEN, AMH, and SAA changes (second-year data) could not be statistically analyzed until two months earlier. Only data from the previous month were obtained. Monthly monitoring, which was carried out for only one year, did not indicate a definite association, as our earlier data did [11]. Therefore, to obtain a straightforward interpretation, long-term monitoring was conducted for 25 months to obtain a total of 2 years of monthly monitoring data. The results of long-term monitoring of the VAR model between the ZEN, AMH, and SAA values and ZEN, AMH, and SAA changes sequentially are shown in Table 3 and Table 4.

The results of long-term monitoring data analysis revealed a significant correlation. Considering the ZEN, AMH, and SAA values comprehensively as shown in Table 3, the AMH level was strongly affected by ZEN in the previous two months (*p* = 0.049) and the AMH level itself in the previous month (*p* = 0.007). The ZEN value of the previous two months showed a negative trend (estimate = −0.042), whereas the AMH value in the previous month showed a positive trend (estimate = 0.41) based on the current AMH value. In other words, a low ZEN value in the previous two months might have caused an increase in the AMH level in the current month, and a low AMH level in the previous month may indicate a low AMH level in this month, and vice versa.

In Table 4, the change (∆) in the AMH value was significantly affected by the previous two months’ ∆ZEN (*p* = 0.016) and the ∆AMH value itself in the previous month (*p* = 0.005). Based on the current ∆ AMH value, the ∆ZEN value for the previous two months (estimate = −0.045) and the ∆AMH value for the previous month (estimate = −0.428) both revealed a negative trend. In other words, a low ∆ZEN value in the previous two months might lead to an increase in the current month’s ∆AMH value, and a low ∆AMH value in the previous month could lead to a high ∆AMH value this month, and vice versa. Further results demonstrated that the changes in the values of ZEN and SAA were strongly influenced by their values from the previous month (*p* = 0.003 and *p* = 0.002, respectively). Both indicated a downward trend (estimate = −0.441 and −0.527), which implies that if the ∆ ZEN or ∆SAA value was high during the previous month, the ∆ ZEN or ∆ SAA value in the following month would decrease, and vice versa. Appendix A, as Appendix A, provides supporting information on the ZEN, AMH, and SAA measurement results.

Finally, based on the findings of the statistical analysis with 2 years of monitoring, we developed formulas to calculate the estimated value of AMH and the change values of AMH, ZEN, and SAA.

Established new formulas:AMH = 0.4 × AMH _Last month_** − 0.04 × ZEN _Last 2 month_* + 434*
ΔAMH = −0.4 × ΔAMH _Last month_** − 0.04 × ΔZEN _Last 2 month_* − 9
ΔZEN = −0.4 × ΔZEN _Last month_** − 34
ΔSAA = −0.5 × ΔSAA _Last month_**

Description of *: *p*-Value = 0.01—<0.05, **: *p*-Value = 0.001—<0.01. R squared (R^2^) in the *AMH*, Δ*AMH*, Δ*ZEN*, and Δ*SAA* formulas is defined as follows: 0.45, 0.37, 0.3, and 0.4.

### 2.2. Calving Intervals as a Reproductive Indicator

The calving interval (days) statistics for 2018, 2019, 2020, 2021, and 2022 are provided in the following order (mean ± SEM): 389.8 ± 35.3; 471.1 ± 33.1; 387.8 ± 15; 408.5 ± 24.6; 368 ± 10. Pre-monitoring data from 2019 were compared with post-monitoring data. Figure 2 shows a decreasing trend from 2019 to 2022, indicating that the calving interval is shortening. The R squared (R^2^) was 0.69 (69%), suggesting that the year variable (urinary ZEN monitoring) had an effect of 69% on the day variable (calving interval). The remaining component (31%) was affected by factors outside this regression equation or variables not investigated. Finally, calving intervals of the cattle herds were compared before and post-ZEN monitoring. Pre-ZEN monitoring was performed in 2019 and post-ZEN monitoring in 2020, 2021, and 2022, with final results after the last year of ZEN monitoring completed in 2022, as shown in Table 5. The number of calving intervals during the post-ZEN monitoring periods ((388.2 ± 10.3) and (368.3 ± 10)) was significantly lower (*p* = 0.007 and *p* = 0.005) than that of the pre-ZEN monitoring period (471.7 ± 33.1).

## 3. Discussion

The primary goal of this feed hygiene monitoring is to raise awareness of the adverse effects of mycotoxins on farmers. Acute exposure to excessive concentrations of mycotoxins in cattle frequently results in well-defined clinical signs such as decreased feed intake or diarrhea. Although ZEN has been shown in laboratory animal studies to cause oxidative stress and cell death [8,21], clinical symptoms of hyperestrogenism in ruminating cows are uncommon [22,23,24]; in our previously reported ZEN-contaminated herds, we were unable to identify the characteristic clinical signs of ZEN [12], but only diarrhea. Sub-chronic and chronic low-dose exposure has been less thoroughly studied; however, it is responsible for decreased performance and potentially the reproductive efficacy of the herd [25]. We used the calving interval as the key indicator for long-term monitoring (two-year surveys) of urinary ZEN to confirm the results on cattle reproductive performance with parameters that might be affected by reproduction. The calving interval for this herd gradually decreased from before monitoring (471.7 days), demonstrating a significant change from the current calving interval (368 days). In addition, the annual herd calving rate increased from 18 to 20 calves. Subsequent long-term urinary ZEN monitoring showed that the concentration of urinary ZEN was low from autumn, when the new rice straw was introduced until early spring of the following year, and then increased in the hot and humid spring and summer. In our previous field study, mycotoxin adsorbent (MA) or oligosaccharide (DFA III) supplementation of ZEN-contaminated feed reduced ZEN absorption in the gut of cattle [13,26]. Therefore, in spring and summer, the administration of MA and DFA III could be an appropriate method to prevent ZEN absorption under low-level chronic ZEN contamination. As pointed out in our previous report [11], urinary ZEN monitoring system is practically useful for herd management and reproductive efficacy in breeding cattle herds.

In our previous study, based on the results of 1 year of monitoring from July 2020 to June 2021, we obtained the tendencies of the relationship between urinary ZEN and AMH concentrations, and those between urinary ZEN and SAA concentrations [11]. In the present study, although no significant relationship was obtained based only on the 2nd-year data, as shown in Table 1 and Table 2, significant co-relationships among the urinary ZEN, AMH, and SAA concentrations in the breeding herd based on the whole 2 years of data were obtained, as shown in Table 3 and Table 4, and we could show the relevance of a clear formula regarding the relationship. The results obtained based on 2 years of data revealed the following: (1) there was a negative correlation between urinary ZEN concentration 2 months earlier and AMH concentration in the current month, (2) there was a negative correlation between the ZEN concentration 1 month ago and that of the current month, (3) the AMH concentration 1 month ago and that in the current month have a negative correlation, and (4) there is a negative correlation between the SAA concentration one month ago and that of the current month. Summarily, it is statistically clear that the instructions and guidance given to farmers based on the results of ZEN-AMH-SAA concentrations being monitored are correctly reflected in the results of the current month. Therefore, it was strongly speculated that feeding hygiene guidance based on our monthly monitoring data for two years led to the above-mentioned improvement in the fertility of the breeding cattle herd. However, new findings regarding AMH concentrations in aging JB cows were obtained from our monitoring. It has been reported that multiparous cows have higher AMH concentrations than primiparous cows throughout the postpartum period, suggesting that plasma AMH concentrations are higher in older cows throughout the postpartum period [27]. The AMH concentrations of fixed cows measured monthly for 2 years (Figure 1) revealed that although the AMH concentration fluctuated (increased or decreased) due to the urinary ZEN concentration, the cow AMH concentration in this herd statistically decreased with aging during the 2-year monitoring period. To the best of our knowledge, this is the first report on AMH decline with age in JB breeding cows, which may indicate a decline in the number of antral follicles in aging cows. Further research using data from additional cows is required to clarify this phenomenon.

## 4. Conclusions

These findings indicate that the urinary ZEN monitoring system may be a valuable practical tool for screening and detecting herd contamination in the field, and acute and/or chronic ZEN contamination in dietary feeds may affect herd productivity and the fertility of breeding cows.

## 5. Materials and Methods

All experiments were conducted in accordance with the Guidelines and Regulations for the Protection of Laboratory Animals and the guidelines of Yamaguchi University (No. 40 of 1995, approved on 27 March 2017). Informed consent was obtained from all participants.

### 5.1. ZEN Monitoring and Reproductive Performance Evaluation

After the detection of very high zearalenone contamination (in urine: >20,250 pg/mL and feed: 1.34 mg/kg) in August 2019, to maintain a similar situation in the following year, the first monitoring of the herd was performed from July 2020 to June 2021 [11]. The monitoring research of the first year was published. To determine the condition of the herd, a second year of monitoring was performed from July 2021 to July 2022 (long-term monitoring, first and second years). Based on the first-year monitoring evaluation, only the ZEN concentrations in urine samples were analyzed in the second year. Five cows from the same feeding and rearing management group were sampled routinely at the beginning of each month, approximately 2 h after the morning feeding. The body weight of the cows was approximately 500 kg, with an age range of 4–7 years. Generally, fresh urine was collected by massaging or stimulating the area surrounding the outer vulva. Approximately 10 mL of fresh urine was collected and stored in a tube, and sample identities were recorded. Fresh urine was stored immediately in a refrigerated sample box and transported to the laboratory. After centrifugation (3500 rpm for 5 min), urine samples were placed in microtubes (approximately 1.5 mL) and stored in a −30 °C freezer until sample analysis was performed. Zearalenone concentrations were performed every two months using ELISA. The detailed analytical methods are described in the next subsection. Calving intervals were referenced and compared annually from 2018 to 2022 as the reproductive records. This study was conducted to verify the efficacy of a monthly ZEN monitoring system. Between 2018 and 2022, there were 10, 18, 19, 20, and 20 parturitions from 24 cows in the herd. A schematic representation of the study design is shown in Figure 3.

### 5.2. Analysis Methods of ZEN, AMH, and SAA

The concentration of ZEN in urine was determined using a commercially available kit (RIDASCREEN^®^ Zearalenon; R-Biopharm AG, Garmstadt, Germany) according to the manufacturer’s instructions with minimal modifications. In brief, a urine sample (0.1 mL:5-fold kit dilution) was added to 3 mL of 50 mM sodium acetate buffer (pH 4.8), and the solution was incubated for 15 h at 37 °C in the presence of 10 μL of β-glucuronidase/arylsulfatase solution. The samples were then placed on a C18 solid-phase extraction (SPE) column (Strata; Phenomenex, Torrance, CA, USA) preconditioned with 3 mL of methanol, followed by 2 mL of 20 mM Tris buffer (pH 8.5) and methanol (80:20). The SPE column was washed with 2 mL of 20 mM Tris buffer (pH 8.5)/methanol (80:20) and 3 mL of 40% methanol before drying by centrifugation (500× *g*) for 10 min. The analytes were gently eluted (flow rate: 15 drops/min) using 1 mL of 80% methanol. Using a centrifugal evaporator, the eluate was evaporated to dryness at 60 °C. The dried residue was redissolved in 50 μL of methanol, followed by 450 μL of sample dilution buffer, properly mixed, and an aliquot of 50 μL was utilized for the ELISA experiment. A RIDA^®^SOFT Win (R-Biopharm, Art. No. Z9999) was used to determine the absorbance at 450 nm using a microplate spectrophotometer to quantify the ZEN concentration in urine samples. Urine creatinine concentrations were assessed using a commercial kit (Sikarikit-S CRE; Kanto Chemical, Tokyo, Japan) and quantified using a 7700 Clinical Analyzer (Hitachi High-Tech, Tokyo, Japan) according to the manufacturer’s instructions. All urine values were given as creatinine ratios (pg/mg creatinine) according to a previous report [12].

To monitor the ovarian antral follicle count (AFC) of the examined cows over the long-term monitoring period, we determined the serum AMH concentration using a bovine AMH ELISA kit (AnshLabs_®_, Webster, TX, USA). In brief, 50 μL of undiluted plasma was added to 50 μL of AMH assay buffer. After 2 h, the cells were incubated and shaken (600–800 rpm) in an orbital microplate shaker. An incubation temperature of 23 ± 2 °C yielded the best results. We aspirated and washed with wash solution (5 times) after incubation, then added the AMH Antibody-Biotin Conjugate-RTU (100 μL), incubated, and shook (600–800 rpm) for 1 h. We washed the solution, added 100 μL of AMH Streptavidin-Enzyme Conjugate-RTU, and incubated for 30 min. Afterward, we washed once more, added 100 μL of TMB chromogen solution, incubated, and shook (600–800 rpm) for 10–12 min. Finally, we added the stopping solution (100 μL) and measured the absorbance of the solution within 20 min with a 450 nm microplate reader. According to the manufacturer’s instructions, the analytical sensitivity and imprecision were 11 pg/mL and 2.92% (coefficient of variation), respectively.

In addition, SAA concentrations were determined during sampling using an automated biochemical analyzer (Pentra C200; HORIBA ABX SAS, Montpellier, France) and a particular SAA reagent for animal serum or plasma (VET-SAA ‘Eiken’ reagent; Eiken Chemical Co. Ltd., Tokyo, Japan). The SAA concentration was calculated using a calibrator-generated standard curve (VET-SAA calibrator set; Eiken Chemical Co., Ltd., Tokyo, Japan).

### 5.3. Feeding Management

The cows in this herd were subjected to the same feed management. Feed is administered to pens as separate forage (79.5%) and concentrate (20.5%) [11]. Forage feeds consist of home-grown rice straw, 12–14 kg (71.8%), and orchard grass, 10 kg per week (7.7%); formula feeds consist of commercially available concentrates, 3 kg (15.4%), and wheat, 0.5–1 kg (5.1%). Orchard grass was provided only once a week. The concentrate administered to cows was a factory-made commercial concentrate tested for mycotoxin contamination. Feed was assessed based on routine monthly monitoring of ZEN contamination in urine. When a high level of zearalenone contamination was identified in urine, we promptly coordinated and discussed the necessity of replacing the feed with veterinarian management and farmers. The presence of ZEN in urine may indicate ZEN contamination in cow feed.

### 5.4. Data Management and Statistical Analysis

Before proceeding with the VAR model, the null hypothesis was evaluated using the Phillips–Perron test. A multivariate time-series study analyzed the monthly estimates of ZEN, AMH, and SAA using the VAR model. The VAR model is a multivariate time-series model that compares current observations of a variable to previous measurements of that variable and other variables in the system and examines the effect of ZEN, AMH, and SAA values from one (lag 1-month) and two months earlier (lag 2-month). In addition, the effects of the changes in ZEN, AMH, and SAA were evaluated by defining the monthly value change (Δ) as the change from the previous month. We examined ΔZEN, AMH, and SAA for their effects on 1- and 2-month lags. Based on the results of the analysis, a *p*-value ≤ 0.05 indicated a significant result. All statistical analyses were performed using the R Project for Statistical Computing, version 4.2.2.

The Mann–Whitney U test (a non-parametric statistical test) was used to determine the normality of the calving interval data as a reproductive record. We performed statistical tests on the 2019 and 2020 calving interval data until 2022 to compare the calving intervals at that time and after the occurrence of contamination. Finally, we compared the differences in the calving intervals when contamination occurred in 2019 and at the end of the monitoring period in 2022. A *p*-value ≤ 0.05 indicates a significantly different result.

## Figures and Tables

**Figure 1 toxins-15-00317-f001:**
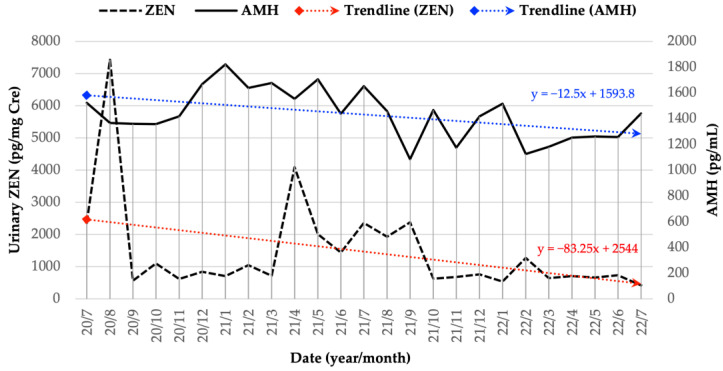
Monthly changes in both urinary ZEN and serum AMH concentrations for 2-year period, mean values of all cows, and linear performance trendlines. ZEN: zearalenone; AMH: anti-Müllerian hormone; y: trend line equation.

**Figure 2 toxins-15-00317-f002:**
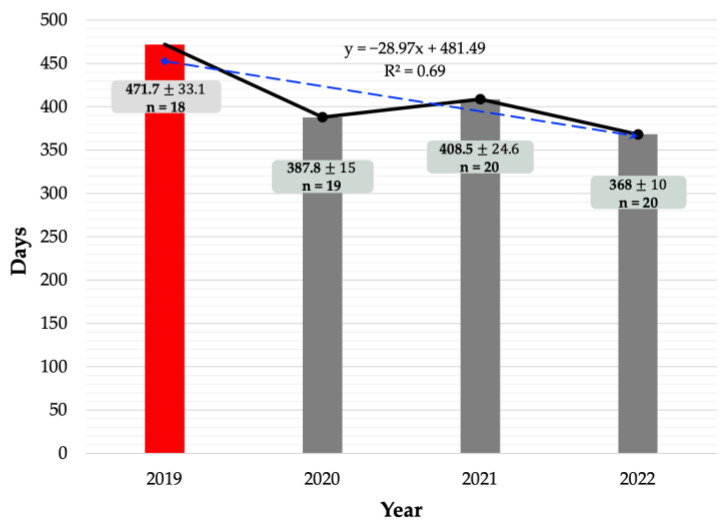
The herd’s calving interval during contamination in 2019 (pre-ZEN monitoring) and after contamination in 2020, 2021, and 2022 (post-ZEN monitoring), mean values, SEM, and annual number of births. SEM: standard error of mean; n: annual number of births; R^2^: R squared; y: trend line equation.

**Figure 3 toxins-15-00317-f003:**
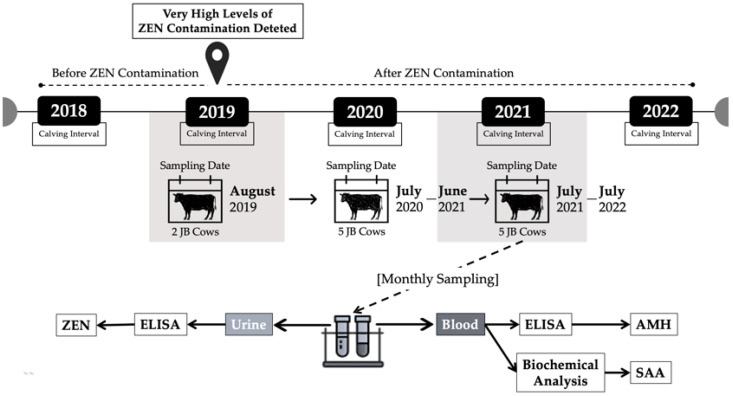
Schematic diagram of experimental design. ZEN: zearalenone; AMH: anti-Müllerian hormone; SAA: serum amyloid A.

**Table 1 toxins-15-00317-t001:** Vector autoregression between ZEN, AMH, and SAA values (2nd-year data).

	ZEN			AMH			SAA		
	Estimate	Std. Error	*p*-Value	Estimate	Std. Error	*p*-Value	Estimate	Std. Error	*p*-Value
ZEN (lag 1-month)	–0.227	0.401	0.611	0.336	0.171	0.145	0.001	0.002	0.238
ZEN (lag 2-month)	0.545	0.229	0.098	–0.247	0.098	0.086	0.001	0.001	0.094
AMH (lag 1-month)	1.116	0.833	0.272	0.214	0.356	0.59	–0.002	0.004	0.646
AMH (lag 2-month)	2.021	0.825	0.092	0.64	0.353	0.167	0.002	0.004	0.71
SAA (lag 1-month)	–61.186	54.531	0.344	5.901	23.325	0.817	–0.603	0.274	0.115
SAA (lag 2-month)	–54.915	39.886	0.262	35.217	17.061	0.131	0.35	0.201	0.179

Std. Error: standard error. (lag 1-month): one month earlier. (lag 2-month): two months earlier.

**Table 2 toxins-15-00317-t002:** Vector autoregression between ZEN, AMH, and SAA change (2nd-year data).

	∆ZEN			∆AMH			∆SAA		
	Estimate	Std. Error	*p*-Value	Estimate	Std. Error	*p*-Value	Estimate	Std. Error	*p*-Value
∆ZEN (lag 1-month)	–0.22	0.463	0.652	0.188	0.177	0.331	–0.001	0.002	0.6
∆AMH (lag 1-month)	0.699	1.198	0.581	–0.142	0.459	0.768	–0.008	0.005	0.184
∆SAA (lag 1-month)	–63.11	62.435	0.351	–9.712	23.925	0.699	–0.455	0.283	0.159

∆: the difference between months or the change in value.

**Table 3 toxins-15-00317-t003:** Vector autoregression between ZEN, AMH, and SAA value (long-term monitoring data).

	ZEN			AMH			SAA		
	Estimate	Std. Error	*p*-Value	Estimate	Std. Error	*p*-Value	Estimate	Std. Error	*p*-Value
ZEN (lag 1-month)	5.75 × 10^−3^	1.30 × 10^−1^	0.965	–0.012	0.021	0.582	–1.48 × 10^−5^	2.66 × 10^−4^	0.956
ZEN (lag 2-month)	–6.12 × 10^−2^	1.27 × 10^−1^	0.632	–0.042	0.021	0.049 *	3.01 × 10^−4^	2.59 × 10^−4^	0.252
AMH (lag 1-month)	9.98 × 10^−1^	8.80 × 10^−1^	0.264	0.41	0.144	0.007 **	–1.47 × 10^−3^	1.80 × 10^−3^	0.42
AMH (lag 2-month)	1.16 × 10^0^	9.21 × 10^−1^	0.216	0.299	0.151	0.054	3.58 × 10^−3^	1.89 × 10^−3^	0.065
SAA (lag 1-month)	1.53 × 10	7.20 × 10	0.833	1.814	11.759	0.878	5.49 × 10^−3^	1.47 × 10^−1^	0.711
SAA (lag 2-month)	–1.02 × 10^2^	7.26 × 10	0.17	12.499	11.859	0.298	2.38 × 10^−1^	1.49 × 10^−1^	0.117

*: *p*-Value = 0.01—< 0.05, **: p-Value = 0.001—< 0.01.

**Table 4 toxins-15-00317-t004:** Vector autoregression between ZEN, AMH, and SAA change (long-term monitoring data).

	∆ZEN			∆AMH			∆SAA		
	Estimate	Std. Error	*p*-Value	Estimate	Std. Error	*p*-Value	Estimate	Std. Error	*p*-Value
∆ZEN (lag 1-month)	–0.441	0.137	0.003 **	–0.003	0.019	0.855	2.3 × 10^−5^	2.7 × 10^−4^	0.933
∆ZEN (lag 2-month)	–0.242	0.129	0.067	–0.045	0.018	0.016 *	3.5 × 10^−4^	2.6 × 10^−4^	0.179
∆AMH (lag 1-month)	1.725	1.049	0.109	–0.428	0.145	0.005 **	–0.002	0.002	0.266
∆AMH (lag 2-month)	1.662	1.064	0.126	–0.199	0.147	0.184	0.002	0.002	0.475
∆SAA (lag 1-month)	27.007	80.606	0.739	–7.057	11.111	0.529	–0.527	0.161	0.002 **
∆SAA (lag 2-month)	–70.452	81.193	0.391	5.148	11.192	0.648	0.057	0.163	0.728

*: *p*-Value = 0.01—< 0.05, **: *p*-Value = 0.001—< 0.01.

**Table 5 toxins-15-00317-t005:** Group statistics on the calving interval between pre- and post-ZEN monitoring.

Calving Interval	*n*	Mean	SEM	*p*-Value	Calving Interval	*n*	Mean	SEM	*p*-Value
Pre-ZEN monitoring (2019)	18	471.7	33.1	0.007 *	Pre-ZEN monitoring (2019)	18	471.7	33.1	0.005 *
Post-ZEN monitoring (2020, 2021, 2022)	59	388.2	10.3		Post-ZEN monitoring (2022)	20	368.3	10	

*n*: number of births (calving); *: significant difference, *p*-Value < 0.05.

## Data Availability

The original contributions presented in the study are included in the article/Appendix A, and further inquiries can be directed to the corresponding author.

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
