# Peer review of "Practical Application of a Urinary Zearalenone Monitoring System for Feed Hygiene Management of a Japanese Black Cattle Breeding Herd—Relevance to Anti-Müllerian Hormone and Serum Amyloid A Clarified from a Two-Year Survey"

_toxins, 2023, doi:10.3390/toxins15050317_

Round 1

Reviewer 1 Report

The authors reported the effects of urinary zearalenone (ZEN) concentration and changes in parameters (AMH and SAA) with time-lag variables and herd fertility (reproductive performance) in a herds of Japanese Black (JB) breeding cattle. Although the originality and novelty are less, the data of urinary ZEN monitoring system can be a valuable practical tool for screening and detecting herd contamination in the field.

I recommend the authors to upload the raw data in the Supplemental material.

Furthermore, in line 20 “fertility f breeding cows” improve it.

In line 143, it is better to indicate the R squared.

Author Response

Thank you very much for your insightful remarks. We completely agree with your suggestions.

  1. Table S1 shows the raw data that has been uploaded.
  2. We have changed it; the right phrase is "fertility of breeding cows."
  3. We specified the R squared on lines 148–149.

Reviewer 2 Report

Please pay attention to express in English languages.

Please pay attention to express in English languages.

Author Response

Thank you very much for your kind advice. In this manuscript, we already used professional editing service from “Editage”.

Reviewer 3 Report

This is a good paper concerning the Application of Urinary Zearalenone Monitoring System for Feed Hygiene Management. I found the work interesting in its exposition and description of methods. I suggest to review :

-the abstract (so as not to start with "here")

- I suggest studying the issue of Feed Hygiene Management in the introduction

- I suggest inserting an image that summarizes the methods and results achieved in a simple way;

Finally, I believe that the bibliography can be deepened with at least 5-6 other entries concerning food hygiene management and the role of ZEN. In this regard, I suggest inserting the following entry:

Llorens Castelló P, Sacco MA, Aquila I, Moltó Cortés JC, Juan García C. Evaluation of Zearalenones and Their Metabolites in Chicken, Pig and Lamb Liver Samples. Toxins (Basel). 2022 Nov 11;14(11):782. doi:10.3390/toxins14110782. PMID: 36422956; PMC ID: PMC9692590.

English is good

Author Response

Thank you very much for your valuable comments. Regarding your suggestion, we revised some parts of our manuscript.

We modified the opening sentence in the abstract section. “In this study, a herd of Japanese Black (JB) breeding cattle with sporadic reproductive disorders was continuously monitored…..”.

Finally, following your suggestions, we inserted one reference (Llorens C.P et al., 2022).

Reviewer 4 Report

The number of bibliographic references should be improved .

Minor editing of English language required

Author Response

According to your suggestion, we added bibliographic references to generate more in-depth content.